# Interplay between Energy Supply and Glutamate Toxicity in the Primary Cortical Culture

**DOI:** 10.3390/biom14050543

**Published:** 2024-04-30

**Authors:** Annette Vaglio-Garro, Andrea Halasz, Ema Nováková, Andreas Sebastian Gasser, Sergejs Zavadskis, Adelheid Weidinger, Andrey V. Kozlov

**Affiliations:** 1Ludwig Boltzmann Institute for Traumatology, The Research Center in Cooperation with AUVA, 1200 Vienna, Austria; annette.vaglio@trauma.lbg.ac.at (A.V.-G.); andrea.halasz@gmx.at (A.H.); ema.novakova25@gmail.com (E.N.); gasser_andreas@gmx.at (A.S.G.); sergejs.zavadskis@trauma.lbg.ac.at (S.Z.); adelheid.weidinger@trauma.lbg.ac.at (A.W.); 2Austrian Cluster for Tissue Regeneration, 1200 Vienna, Austria

**Keywords:** primary cortical cultures, cellular viability, mitochondria membrane potential, lactate dehydrogenase activity, lactate shuttle, glutamate toxicity

## Abstract

Limited substrate availability because of the blood–brain barrier (BBB) has made the brain develop specific molecular mechanisms to survive, using lactate synthesized by astrocytes as a source of energy in neurons. To understand if lactate improves cellular viability and susceptibility to glutamate toxicity, primary cortical cells were incubated in glucose- or lactate-containing media and toxic concentrations of glutamate for 24 h. Cell death was determined by immunostaining and lactate dehydrogenase (LDH) release. Mitochondrial membrane potential and nitric oxide (NO) levels were measured using Tetramethylrhodamine, methyl ester (TMRM) and 4-Amino-5-Methylamino-2′,7′-Difluorofluorescein Diacetate (DAF-FM) live staining, respectively. LDH activity was quantified in single cells in the presence of lactate (LDH substrate) and oxamate (LDH inhibitor). Nuclei of cells were stained with DAPI and neurons with MAP2. Based on the distance between neurons and glial cells, they were classified as linked (<10 µm) and non-linked (>10 µm) neurons. Lactate increased cell death rate and the mean value of endogenous NO levels compared to glucose incubations. Mitochondrial membrane potential was lower in the cells cultured with lactate, but this effect was reversed when glutamate was added to the lactate medium. LDH activity was higher in linked neurons compared to non-linked neurons, supporting the hypothesis of the existence of the lactate shuttle between astrocytes and at least a portion of neurons. In conclusion, glucose or lactate can equally preserve primary cortical neurons, but those neurons having a low level of LDH activity and incubated with lactate cannot cover high energetic demand solely with lactate and become more susceptible to glutamate toxicity.

## 1. Introduction

Neuronal death upon neurodegenerative disorders and brain injury are associated with excessive production of NO and reactive oxygen species (ROS) [1], mitochondrial dysfunction [2,3] and glutamate toxicity [4,5]. Excessive production of ROS is associated with electron leak from the mitochondrial respiratory chain, while the excessive NO generation is linked to the activation of neuronal nitric oxide synthase (nNOS) [6]. The accumulation of extracellular glutamate is preceded by several upstream molecular events, for example, low oxoglutarate dehydrogenase complex (OGDHC) activity in mitochondria, activation of inducible nitric oxide synthase (iNOS), and spontaneous release of glutamate [7], which is controlled by mitochondria [3]. However, causal interaction between these three factors is still not completely understood. 

In the brain, energy supply to neurons has a specific feature of restrictive diffusion due to the BBB. It controls the entrance of macromolecules to the brain and protects the central nervous system (CNS) from neurotoxins [8,9]. One of the proposed mechanisms to provide neurons with enough energy, which covers their high energetic demand required for neurotransmission [10], is the astrocytes–neurons lactate shuttle. This shuttle was described by Pellerin and Magistretti in 1994 [11]. It features an important molecular mechanism for astrocyte-mediated energy supply to neurons compensating their inability to obtain substrates directly from the bloodstream [12,13,14,15,16,17].

Another important interaction between neurons and astrocytes in the brain, which contributes to overall homeostasis, is the glutamate–glutamine cycle. It serves for signal transmission and prevents toxic effects of glutamate [7,18]. Astrocytes collect excessive glutamate that is released during synaptic transmission and convert it into non-toxic glutamine by the glutamine synthetase [18,19,20,21,22]. Glutamine is subsequently transported into neurons to turn back into glutamate by the phosphate-activated glutaminase [18,20,23]. Eventually, glutamate is packed in transmission vesicles or used in the mitochondrial tricarboxylic acid (TCA) cycle to produce energy [3,7,24]. 

One of the advanced techniques often used to understand the pathogenesis of neural disorders is primary brain cell cultures [3,25]. This is the most useful and widely used model to study pathophysiological impacts of glutamatergic receptors like N-methyl-D-aspartate receptor (NMDAR) and α-amino-3-hydroxy-5-methyl-4-isoxazole propionic acid receptors (AMPARs), which are well characterized and expressed in this type of cell [26,27]. 

However, it is important to note that neuronal culture cannot completely recapitulate morphological structures of the brain, and it does not contain the BBB. Subsequently, *in vitro* models require components that are provided with the incubation medium, meaning all substrates are available for the neurons without restrictions. This raises questions whether all parts of the lactate shuttle are operational in this model.

The aim of this study was to clarify whether incubation of primary cortical cells with glucose or lactate, as energy sources affects the cellular viability and susceptibility to glutamate toxicity (Figure 1). This knowledge is essential for accurate interpretation of the results obtained in neuronal cell culture experiments. 

## 2. Materials and Methods

### 2.1. Isolation of Primary Cells from Brain Cortex

Pregnant Sprague-Dawley rats were provided by Janvier Labs, France. The protocol was based on [28] with some modifications. Neonatal animals were euthanized by decapitation in full accordance with all rules of the Austrian animal and experimental protection law, which implement European regulations. The responsible animal welfare oversight body was the “Institutional Review Board (Tierschutzgremium) of the Ludwig Boltzmann Institute of Traumatology”. Brains were extracted to dissect the cortex in ice-cold buffer which contained (in mM) the following: 137 NaCl (Sigma-Aldrich, Darmstadt, Germany), 5.4 KCl (Sigma-Aldrich, Darmstadt, Germany), 1.1 Na_2_HPO_4_ (Sigma-Aldrich, Darmstadt, Germany), 1.1 KH_2_PO_4_ (Sigma-Aldrich, Darmstadt, Germany), 6.1 glucose (Sigma-Aldrich, Darmstadt, Germany), and 1 kynurenic acid (Sigma-Aldrich, Darmstadt, Germany), with pH adjusted to 7.3 using NaOH (Sigma-Aldrich, Darmstadt, Germany). Primary co-cultures of cortical neurons and glial cells were prepared after enzymatic digestion of brain tissue for 17 min at 37 °C and 5% CO_2_ with papain (Worthington Biochemical Corporation, Solana Beach, CA, USA) at a concentration of 25 units/mL in L-15 Leibovitz medium (Sigma-Aldrich, Darmstadt, Germany) supplemented with 2 mM kynurenic acid (Sigma-Aldrich, Darmstadt, Germany) and mechanical dissociation with Pasteur pipettes (trituration). The culture medium used was based on Dulbecco’s Modified Eagle’s Medium (DMEM)-high glucose (D5796, Sigma-Aldrich, Darmstadt, Germany) supplemented with 10% heat-inhibited fetal bovine serum (FBS, Biowest, Nuaillé, France), 12.5 nM progesterone (Sigma-Aldrich, Darmstadt, Germany), 112.5 μM putrescine dihydrochloride (Sigma-Aldrich, Darmstadt, Germany), 6.25 μg/mL insulin, 6.25 μg/mL transferrin, and 6.25 ng/mL sodium selenite (added from a 400-times concentrated stock solution of Insulin-Transferrin-Sodium Selenite Supplement, Roche Austria GmbH, Vienna, Austria). The medium also contained 25,000 U/L penicillin and 25 mg/L streptomycin (Sigma-Aldrich, Darmstadt, Germany), which was added from a 400-times concentrated stock. After the final stage of single-cell suspension, the cells were seeded at a density of 400,000 cells into 6-well plates with glass coverslips inside, previously coated with Poly-D-Lysine (Sigma-Aldrich, Darmstadt, Germany). The initial medium was exchanged after 24 h with an identical medium without antibiotics. One micromolar dose of cytosine arabinoside (Ara-C, Sigma-Aldrich, Darmstadt, Germany) was added at day 5 after the preparation to reduce the proliferation of non-neuronal cells. The cells were kept in culture for 6–7 days at 37 °C, 5% CO_2_, and 100% humidity.

### 2.2. Glutamate Toxicity Curve

On day 6, the cells in culture were treated with different concentrations of glutamate (0 µM, 50 µM, 150 µM, 300 µM, 1000 µM, and 50,000 µM, Sigma-Aldrich, Darmstadt, Germany). To add the treatment, half of the medium was removed from each well. Double the amount of the desired final concentration of glutamate was added to the medium, then it was mixed and put back again into the wells to prevent the removal of the neuronal growing factors that were already established in the wells. The cells were incubated with the treatment for 24 h at 37 °C, 5% CO_2_, and 100% humidity. One hundred intact nuclei (Figure 2) of three fields per sample were counted based only on the DAPI signal (specific marker to label nuclei of all cells—glia and neurons). Viable neurons were identified and included only if the cells were positive for MAP2 (specific biomarker for neurons, anti-Alexa594, Proteintech Germany GmbH, Planegg-Martinsried, Germany). The data were normalized to percentage by using the control as 100%. Imaging was performed on a Nikon microscope with a DS-FI3 camera (Nikon, Tokyo, Japan) with a 10× objective magnification. The images were acquired in NIS-Elements software (version 5.21.00, Nikon, Tokyo, Japan). Images were processed in ImageJ (Version 2.14, National Institutes of Health, NIH, Bethesda, MD, USA).

### 2.3. Immunostaining

After removing the medium from the cells, they were washed with PBS 1× (Sigma-Aldrich, Darmstadt, Germany), fixed with 4% formaldehyde (VWR Chemicals BDH, Radnor, PA, USA) for 10 min, washed with PBS 1X (Sigma-Aldrich, Darmstadt, Germany), and permeabilized with 0.5% Triton (Sigma-Aldrich, Darmstadt, Germany) for 5 min. The cells were incubated at 4 °C with 8% Normal Goat Serum (NGS, previously diluted in PBS 1×, Histoprime BIOZOL, Eching, Germany). Afterwards, 594-conjugated MAP2 Rabbit PolyAb-Antibody (1:250, diluted in 8% NGS; Proteintech Germany GmbH, Planegg-Martinsried, Germany) was added to the cells and incubated overnight at 4 °C in darkness. On the next day, the cells were washed with PBS 1× (Sigma-Aldrich, Darmstadt, Germany), dried for one hour at room temperature in darkness, and mounted with Prolong Gold Antifade DAPI (ThermoFisher Scientific, Waltham, MA, USA) on top of glass microscope slides. Imaging was performed on a Nikon microscope with a DS-FI3 camera (Nikon, Tokyo, Japan) with a 10× objective magnification. The images were acquired in NIS-Elements software (version 5.21.00, Nikon, Tokyo, Japan). Images were processed in ImageJ (Version 2.14, National Institutes of Health, NIH, Bethesda, MD, USA). Glia cells were stained with 488-conjugated GFAP Mouse McAb-Antibody (1:50, diluted in 8% NGS; Proteintech Germany GmbH, Planegg-Martinsried, Germany), and data are shown in the Appendix A.

### 2.4. Colorimetric Enzymatic Activity Assay

Primary cortical cells were washed with PBS 1× (Sigma-Aldrich, Darmstadt, Germany), fixed with 4 % formaldehyde (VWR Chemicals BDH, Radnor, PA, USA) for 10 min, washed with PBS 1× (Sigma-Aldrich, Darmstadt, Germany), and permeabilized with 40 µM Digitonin (Sigma-Aldrich, Darmstadt, Germany) for 10 min at 37 °C. After an additional wash with PBS 1× (Sigma-Aldrich, Darmstadt, Germany), different compounds were added to the cells: lactate (substrate, Sigma-Aldrich, Darmstadt, Germany), lactate + oxamate (substrate + inhibitor, Sigma-Aldrich, Darmstadt, Germany + ThermoFisher Scientific, Waltham, MA, USA), and no treatment (negative control). We based our protocol on [29] with some modifications. Each compound was dissolved in Tris-Maleate buffer (0.1 M Tris-Maleate (Sigma-Aldrich, Darmstadt, Germany) buffer pH 7.5 with 10% polyvinyl alcohol (Sigma-Aldrich, Darmstadt, Germany)), which was prepared and dissolved at 85 °C. Four hundred and fifty micromolar of methoxyphenamine methosulfate (transfers electrons, Sigma-Aldrich, Darmstadt, Germany), 5 mM sodium azide (inhibitor of mitochondrial respiration, Sigma-Aldrich, Darmstadt, Germany), 5 mM nitroblue tetrazolium chloride (color indicator, Sigma-Aldrich, Darmstadt, Germany), and 3 mM β-nicotinamide adenine dinucleotide hydrate (coenzyme of LDH enzyme, Sigma-Aldrich, Darmstadt, Germany) were added to the Tris-Maleate buffer. The nitroblue tetrazolium chloride and methoxyphenanzine methosulfate were pre-dissolved in a mixture of 50% ethanol (Carl Roth, Karlsruhe, Germany) and 50% dimethyl formaldehyde (Sigma-Aldrich, Darmstadt, Germany). The negative control did not contain lactate or oxamate. For the substrate, 150 mM sodium lactate (Sigma-Aldrich, Darmstadt, Germany) was added. For the substrate + inhibitor reaction, 150 mM of sodium lactate and 200 mM of sodium oxamate (ThermoFisher Scientific, Waltham, MA, USA) were added. These three different compounds were added per duplicate into the 6-well-plates. The cells were incubated for 20 min at room temperature in 110 rpm agitation (VWR advanced digital shaker). The enzymatic reaction was stopped by washing the samples with 70 °C PBS 1× (Sigma-Aldrich, Darmstadt, Germany) for five times until the assay medium was completely removed, followed by two washes with 4 °C PBS 1× (Sigma-Aldrich, Darmstadt, Germany). For the immunostaining, the coverslips were blocked with 8% Normal Goat Serum (previously diluted in PBS 1×, Histoprime BIOZOL) overnight at 4 °C. The next day, the cells were incubated with 594-conjugated MAP2 Rabbit PolyAb-Antibody (1:250, diluted in 8 % NGS; Proteintech Germany GmbH, Planegg-Martinsried, Germany) for 2 h at 4 °C. Afterwards, the coverslips were washed with PBS 1× (Sigma-Aldrich, Darmstadt, Germany), dried for one hour, and mounted with Prolong Gold Antifade DAPI (ThermoFisher Scientific, Waltham, MA, USA) onto a glass microscope. Imaging was performed on a Nikon microscope with a DS-FI3 camera (Nikon, Tokyo, Japan) with a 10× objective magnification. The images were acquired in NIS-Elements software (version 5.21.00, Nikon, Tokyo, Japan). Images were processed in ImageJ (Version 2.14, National Institutes of Health, NIH, Bethesda, MD, USA). 

### 2.5. Cell Culture with Different Substrates and Toxic Concentrations of Glutamate

Six days *in vitro* of isolated cortical primary cells were incubated in DMEM-high glucose (D5796, Sigma-Aldrich, Darmstadt, Germany) supplemented with 10% heat-inhibited FBS (Biowest, Nuaillé, France), 12.5 nM progesterone (Sigma-Aldrich, Darmstadt, Germany), 112.5 μM putrescine dihydrochloride (Sigma-Aldrich, Darmstadt, Germany), 6.25 μg/mL insulin, 6.25 μg/mL transferrin, and 6.25 ng/mL sodium selenite (added from a 400-times concentrated stock solution of Insulin-Transferrin-Sodium Selenite Supplement, Roche Austria GmbH, Vienna, Austria) or in DMEM-free glucose Medium (11966025, Gibco/ThermoFisher Scientific, Waltham, MA, USA) supplemented with 10% heat-inhibited FBS (Biowest, Nuaillé, France), 12.5 nM progesterone (Sigma-Aldrich, Darmstadt, Germany), 112.5 μM putrescine dihydrochloride (Sigma-Aldrich, Darmstadt, Germany), 6.25 μg/mL insulin, 6.25 μg/mL transferrin, 6.25 ng/mL sodium selenite (added from a 400-times concentrated stock solution of Insulin-Transferrin-Sodium Selenite Supplement, Roche Austria GmbH, Vienna, Austria), and 10 mM of sodium lactate (Sigma-Aldrich, Darmstadt, Germany). Each batch of cells was treated with or without 50,000 μM of glutamate (Sigma-Aldrich, Darmstadt, Germany) for 24 h at 37 °C, 5% CO_2_, and 100% humidity. After incubation, the supernatants were collected from one batch of cells, centrifuged (PCV-2400 combined centrifuge vortex mixer) at 2800 rpm for 5 min at room temperature, and used to measure LDH release. The cells were fixed and treated as previously described in the immunostaining and colorimetric enzymatic activity assay sections. Imaging was performed on a Nikon microscope with a DS-FI3 camera (Nikon, Tokyo, Japan) with a 10× objective magnification. The images were acquired in NIS-Elements software (version 5.21.00, Nikon, Tokyo, Japan). Images were processed in ImageJ (Version 2.14, National Institutes of Health, NIH, Bethesda, MD, USA). 

### 2.6. LDH Release Assay

LDH release was measured as [3], by mixing cell culture supernatant (30 μL) with 100 μL LDH assay reagent containing 110 mM lactic acid (Sigma-Aldrich, Darmstadt, Germany), 1350 μM nicotinamide adenine dinucleotide (NAD+, Sigma-Aldrich, Darmstadt, Germany), 290 μM N-methylphenazonium methyl sulfate (PMS, Sigma-Aldrich, Darmstadt, Germany), 685 μM 2-(4-iodophenyl)-3-(4-nitrophenyl)-5-phenyl-2H-tetrazolium chloride (INT, Sigma-Aldrich, Darmstadt, Germany), and 200 mM Tris (pH 8.2, Sigma-Aldrich, Darmstadt, Germany). The kinetics of changes in absorbance (492 nm) were recorded for 45 min (kinetic LDH assay). LDH activity values were determined as maximal velocity of NADH formation (mOD/min). Data were divided between glucose or lactate averages, respectively, to obtain folds values.

### 2.7. Mitochondrial Membrane Potential and Intracellular Concentrations of Nitric Oxide

Primary cortical cells were washed twice with PBS 1× (Sigma-Aldrich, Darmstadt, Germany) for 5 min each. The cells were incubated with 50 nM of Tetramethylrhodamine, methyl ester (TMRM, Thermo Fisher Scientific, Waltham, MA, USA) and 10 μM of 4-Amino-5-Methylamino-2′,7′-Difluorofluorescein Diacetate (DAF-FM, Thermo Fisher Scientific, Waltham, MA, USA) staining solution for 30 min at 37 °C, 5% CO_2_, and 100% humidity. The cells were washed one time with PBS 1× (Sigma-Aldrich, Darmstadt, Germany). The imaging was performed by a Zeiss LSM510 Meta laser scanning confocal microscope (Zeiss, Jena, Germany), with ex/em of 543/570 nm, to detect TMRM and ex/em of 488/515 nm for DAF-FM detection with a 10× objective magnification. The cells were screened into neurons and glial cells by measuring the diameter of the cellular body. Cells with diameter smaller than 10 µm (area lower than 100 µm^2^) were classified as neurons, and larger-diameter cellular bodies were assigned as glial cells (see Appendix A). The analysis of fluorescent signals was measured as mean gray intensity for individual cells in ImageJ (Version 2.14, National Institutes of Health, NIH, Bethesda, MD, USA). Cells in division, damaged, or dead were excluded from the analysis (Figure 2). The size of the neurons and glial cells were determined based on the images belonging to immunofluorescence where only neurons were positive for MAP2; the protocol is explained in Appendix A.

### 2.8. Linked and Non-Linked Neurons

The distance between neurons and other cells was measured in micrometers. The neurons that were located alone or farther than 10 µm from other cells were classified as non-linked neurons. Neurons in close distance (less than 10 µm) to other cells were classified as linked neurons. The average of linked neurons was used as 100% to normalize all the data into percentages.

### 2.9. Statistic Analysis

Data distribution was checked by a Shapiro–Wilk test, and outliers were excluded based on the ROUT (Q = 1%) test. Calculations were performed by using GraphPad prism version 10 for Windows, GraphPad Software (Version 10.1.2, San Diego, CA, USA). The statistical test and number of biological and technical replicates used per analysis were stated in the legend of the figures, and all data were expressed as average ± SEM.

## 3. Results

First, we examined the effect of different concentrations of glutamate on neuron morphology and viability by staining cells with MAP2, a neuronal marker. A summary of the experimental design is shown in Figure 3A. Figure 3B shows that in the absence of glutamate, neurons have long dendrites and rounded bodies [30]. Increasing concentrations of glutamate reduced the size of the dendrites and the diameter of cellular bodies (Figure 3C–G); simultaneously, we observed a drop in neuronal viability. The viability was quantified by counting MAP2-positive cells (Figure 3H) and LDH release (Figure 3I). Both methods show 10-fold reduction in the neuronal viability upon 50,000 µM glutamate treatment. Conditions with up to 300 µM of glutamate show a significant decrease in neuronal viability in comparison with the control or with 50 µM of glutamate. When we compared 150 µM of glutamate with 1000 µM and 50,000 µM of glutamate, we observed a significant difference in neuronal viability. Between 300 µM, only 50,000 µM of glutamate shows a significant decrease in neuronal viability (Figure 3H).

We then tested the effect of glucose and lactate as the energy source for neurons in the presence and in the absence of glutamate (Figure 4). In the absence of glutamate, we did not observe any changes in the cell death rate; however, in the presence of glutamate, cells incubated with lactate manifested higher death rate compared to those incubated with glucose (Figure 4B). Since neuronal death is often associated with elevated NO generation, we determined whether there is a difference in NO levels between the cells incubated with glucose and lactate in the presence and in the absence of glutamate (Figure 5).

To analyze these data in live cells, neurons were separated from the other cells based on the size of their cellular body area calculated by using the images wherein neurons were stained with MAP2 (see Appendix A). We observed that in all cells (neurons + glial cells), there is a strong but not significant trend to increase the NO levels in the presence of lactate (Figure 5A–L, quantification Figure 5M–P). This trend was more pronounced, reaching a nearly significant level (*p* = 0.0515) in neurons (Figure 5O). Another parameter, which is often associated with neuronal death, is mitochondrial dysfunction. To characterize mitochondria, we determined mitochondrial membrane potential (MMP) in live cells. The images obtained in live cells are shown in Figure 6A–L. We did not observe a significant difference in the total cell population (Figure 6M,N). In contrast, there was a remarkable difference in the neurons (Figure 6O,P). MMP was significantly lower in the presence of lactate compared to cells incubated with glucose (Figure 6O), while in the presence of glutamate, we observed opposite changes, where MMP was higher in cells incubated with lactate (Figure 6P). 

The data on MMP suggest that cells incubated with glucose and lactate have differently regulated bioenergetics status. Active LDH is the obligatory prerequisite for energy supply to neurons via lactate. Considering this fact, we determined the activity of LDH in single neurons. For this, we adapt the approach to determine single-cell enzymatic activities reported in [29] for our experimental model. 

As shown in Figure 7, LDH activity is possible to detect in single fixed cells (neurons or glial cells). Total numbers of cells were identified by staining the nuclei of the cells (Figure 7A,E,I), neurons were marked by a specific neuronal-cytoplasmic marker MAP2 (Figure 7B,F,J), the green signal in the cytoplasm of the cells signifies the LDH activity per cell (Figure 7C,G,K), and the combination of all the signals can be observed in Figure 7D,H,L. In Figure 7A–D, we detect LDH activity in each individual cell when the substrate (lactate) was added to the detection reaction (Figure 7D). In the presence of oxamate (Figure 7E–H), a competitive inhibitor of LDH enzyme, the efficiency of the enzyme was substantially lower (compared with the image Figure 7D), showing a low number of positive cells for LDH activity (Figure 7H). In Figure 7I–L, we do not detect LDH activity. There is no LDH activity in our system in the absence of substrate (negative control, Figure 7L). 

After identifying the activity and localization of LDH enzyme in our cellular model, we wanted to deepen our understanding of the molecular mechanism which lies behind our observations. Many papers explain how the lactate shuttle allows the movement of lactate from astrocytes to neurons, mainly in the absence of glucose, to supply the neurons with a suitable substrate for efficient ATP production [31,32]. 

In Figure 8, we detected LDH activity in individual fixed neurons. Cellular nuclei were identified by DAPI staining (Figure 8A,E), MAP2 was used to specifically identify the neurons (Figure 8B,F), LDH activity per cell was measured via a redox reaction with nitro blue tetrazolium chloride (Figure 8C,G), and in the merged images in Figure 8D,H, all signals were combined. Figure 8A–D show that neurons in close distance to glial cells (<10 µm) have the highest LDH activity (Figure 8C). These neurons were named by our group as linked neurons (Figure 8B,D). On the other hand, the neurons that are far away from glial cells at larger distances than >10 µm (Figure 8E–H, non-linked neurons) present a lower LDH activity (Figure 8G,H) in comparison to Figure 8C,D. In the merged figures, it is possible to confirm the colocalization between the LDH-positive signal and MAP2 in linked (Figure 8D) and in non-linked (Figure 8H) neurons. The result was quantified in terms of percentage LDH activity in linked and non-linked neurons. The linked neurons have a statistically significantly higher LDH activity compared to the non-linked neurons (Figure 8I). Comparison between live linked and non-linked neurons cultured in glucose or lactate medium is in Appendix A. 

The low or the absence of LDH activity in non-linked neurons suggests that they may be more susceptible to death upon incubation with lactate. To address this question, we performed additional experiments to compare the loss of linked and non-linked neurons upon treatment with lactate and glucose. The results of this experiment are shown in the Appendix A. We observe that the ratio non-linked/linked neurons is higher in cells incubated with glucose compared to those incubated with lactate, which supports our assumption. 

## 4. Discussion

Primary cortical cell culture is a widely used model to study pathophysiological mechanisms occurring in brain disorders with the particular emphasis of glutamate toxicity. *In vivo*, due to the BBB, the direct transport of glucose to neurons is limited under certain circumstances, and this becomes a substantial portion of energy through the lactate shuttle from astrocytes [33,34,35,36,37,38]. In contrast to *in vivo* reality, primary brain cultures are cultivated in the presence of sufficient amounts of glucose medium and likely do not require being supplied by lactate. Here, we examined primary cortical cultures, isolated from newborn rats, incubated with either glucose or lactate and their response to toxic glutamate concentrations.

We did not observe any difference in the cell death rate between the cells which were incubating with glucose or lactate (Figure 3). These data are in line with previous publications [39]. The authors quantified relative viability of primary cortical neurons treated with 20 mmol/L of glucose and 10 mmol/L of lactate and did not report any difference [39]. However, the cells incubated with lactate exhibited increased levels of NO and iNOS. Similar results were reported in the SH-SY5Y neuronal cell line and in primary cortical astrocytes, following incubation with 10 mmol/L of lactate for 24 h [40]. The correlation between nitric oxide production and LDH release has already been reported [41], suggesting the activation of an NO-mediated mechanism of cell death. Here, we observed that NO levels had a strong trend to increase in the presence of lactate (Figure 5). 

The experiments related to the effect of toxic glutamate concentration showed that cells incubated with lactate were more susceptible to glutamate toxicity. The death rate in response to glutamate was higher in cells incubated with lactate compared with cells incubated with glucose (Figure 3). Similar results were reported by Llorente-Folch et al. in cortical neuronal cells from mice [42]. Since mitochondrial dysfunction contributes to neuronal death, we determined MMP in vital neuronal cells and its response to toxic concentrations of glutamate. As expected, MMP decreased upon treatment with glutamate in the cells incubated with glucose (Figure 6). This is in line with already described mitochondrial mechanisms underlying glutamate toxicity [2]. In contrast, we observed an increase in MMP upon glutamate treatment in the cells incubated with lactate [42]. We assume that this phenomenon is related to lactate metabolism. 

It is important to mention that there are some reports wherein is has been shown that increased concentrations of glutamate can trigger lactate production [43] as well as potentiates calcium response through lactate [44]. This suggests that there are other biochemical side reactions, which could occur in our model that can also contribute in a small part to our evidenced cell death. 

The delivery of energy to neurons via lactate shuttle requires the presence of active LDH enzyme in neurons and glial cells. Since we supply lactate to the cells in the medium, only the LDH enzyme located in the neurons is critical. To examine the LDH activity in neurons, we adopted a previously developed method [29] for our experimental model, which enabled us to determine LDH activity in single cells. We observed that all glial cells without exception manifested high LDH activity (Figure 7). In this study, we did not identify all glial cells occurring in our primary culture, but it is commonly accepted that astrocytes predominantly supply neurons with lactate. 

Determination of the LDH activity in single neurons revealed that they are heterogeneous in terms of LDH activity (Figure 8). Heterogeneity in LDH activity in single cells was already reported in lymphocytes [45]. We have shown that only neurons located close to glial cells (linked neurons) have a well-detectable LDH activity, while neurons located far from glial cells do not exhibit detectable LDH activity (Figure 8). High activity of LDH in linked neurons observed in our model suggests the presence of astrocyte–neuron lactate shuttle, which is described elsewhere [46,47]. 

We do not know the reason for this heterogeneity in the neuronal population, but we assume that it could be a characteristic feature for cell culture. This finding suggests that in a primary cortical culture, not all neurons are equally supplied by energy in the presence of lactate (Appendix A). We speculate that those neurons with low LDH activity may develop a predisposition to death, e.g., via elevated production of NO by nNOS and the energy supply is shut down. Indeed, it has previously been suggested that glutamate excitotoxicity can be mediated by NO [48,49]. Thus, elevated NO levels can predispose cells to glutamate toxicity, although the death pathway is not executed in the absence of glutamate [6]. 

This also may explain the paradoxical reaction of MMP in response to glutamate in the cells incubated with lactate. It can be due to an additional electron supply to the TCA-cycle from glutamate. However, this improvement of mitochondrial function evidently does not rescue the cells, because death mechanisms mediated by glutamate are predominant over partial recovery of mitochondrial function. 

There are different reasons for the heterogeneity of LDH activity in neurons. Presumably, contact with astrocytes is an obligatory prerequisite for LDH activity in neurons. Alternatively, it can be due to the post-translational regulation of LDH activity. Currently, this question remains unclear and demands further research. 

## 5. Conclusions

The evidence collected in this study suggests that the experiments involving energy supply from lactate performed in the primary neuronal culture should be interpreted with caution, as these results may arise due to pathways which are not operational *in vivo*. Primary cortical neurons can be equally well preserved if incubated with either glucose or lactate. However, cells incubated with lactate are more susceptible to glutamate toxicity. Our data suggest that this is due to the low LDH activity in a portion of neurons that are not in contact with glial cells and cannot cover the increased energy demand solely from lactate.

## Figures and Tables

**Figure 1 biomolecules-14-00543-f001:**
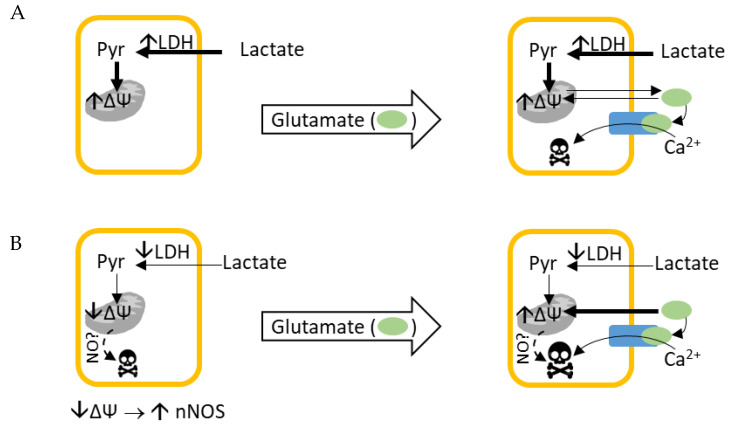
Graphical abstract of our working hypothesis. (**A**) Primary neurons linked to glial cells have high LDH activity, do not require glutamate for energetical purposes, and are less susceptible to glutamate toxicity. (**B**) Primary neurons not linked to glial cells have low LDH activity, require additional substrate (e.g., glutamate) for energetical purposes, and are more susceptible to glutamate excitotoxicity.

**Figure 2 biomolecules-14-00543-f002:**
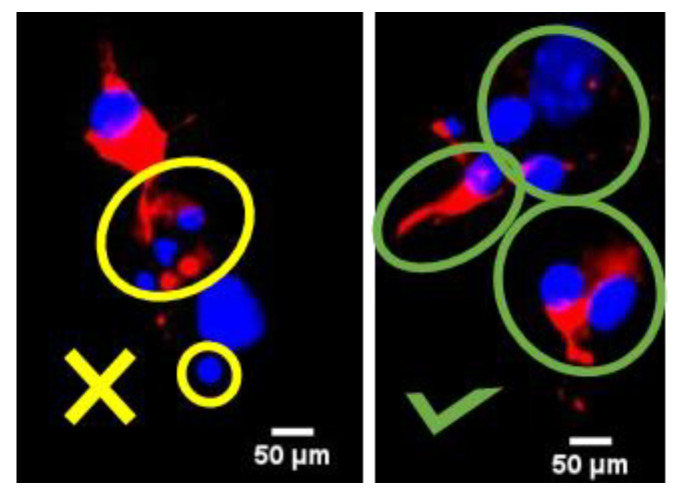
Image analysis included categorization of the nuclei based on their shape. The yellow circles on the left image demonstrate fragmented and damaged nuclei or debris, which were excluded from the analysis. On the right image, green circles outline intact nuclei, which were included in the analysis. Red: MAP2 marker to detect microtubules in the neurons; blue: DAPI. Scale bar = 50 µm.

**Figure 3 biomolecules-14-00543-f003:**
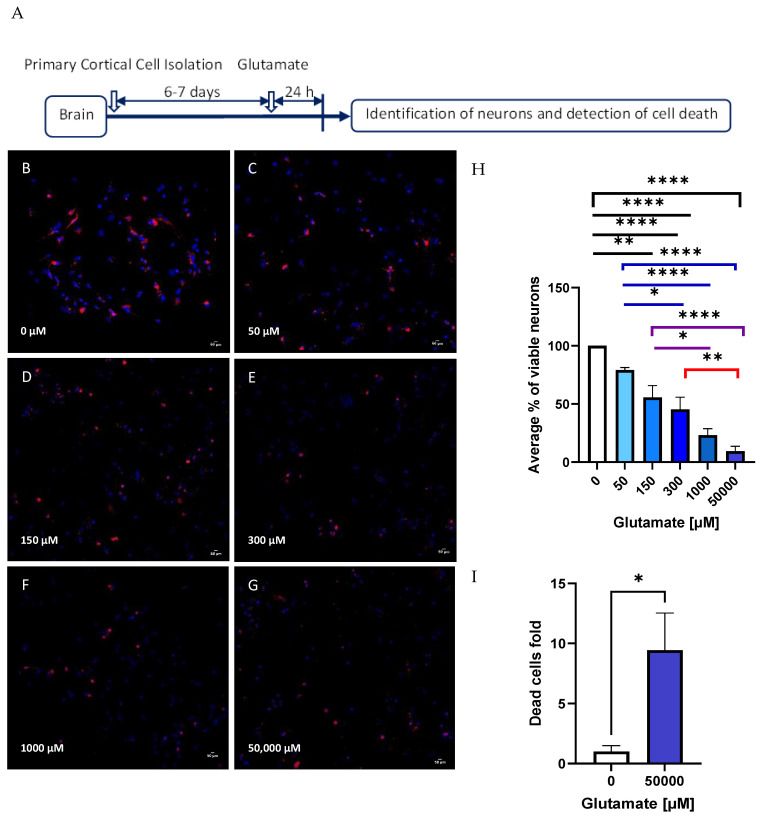
Primary cortical cells isolated from Sprague-Dawley rats. (**A**) Experimental design of cell isolation and treatment with glutamate. The cells were treated with different concentrations of glutamate: (**B**) control (no treatment), (**C**) 50 µM, (**D**) 150 µM, (**E**) 300 µM, (**F**) 1000 µM, and (**G**) 50,000 µM. (**H**) Shows the quantification of percentage of viable neurons (detected by immunostaining method) after the treatment. Higher concentrations of glutamate decrease the percentage of viable neurons. (**I**) The same result was confirmed by an enzymatic method used to detect dead cells (LDH release). Both methods show a reduction in neurons in 10-folds when the primary cortical cells were treated with 50,000 µM in comparison to the control. Treatment with 50,000 µM of glutamate killed 90% of neurons. Data are presented as a mean ± SEM. n = 5–7 independent biological samples with 2–3 technical replicates. Statistical significance was determined using one-way ANOVA and Tukey’s multiple comparison tests. * *p* < 0.05; ** *p* < 0.01; **** *p* < 0.0001. Red: MAP2 marker to detect microtubules in the neurons; blue: DAPI. Scale bar = 50 µm.

**Figure 4 biomolecules-14-00543-f004:**
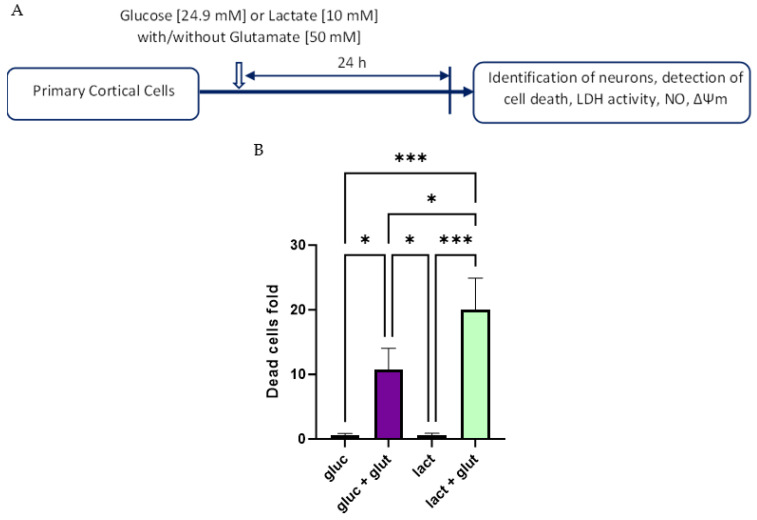
(**A**) Experimental design of cells cultured with specific media. The cells were isolated and cultured for 6–7 days. On the last day of culture, the medium was exchanged for glucose standard medium [24.9 mM] or glucose-free medium with lactate [10 mM] and glutamate [50 mM]. (**B**) Dead primary cortical cells, isolated from Sprague-Dawley rats, after adding different treatments: glucose, glucose + glutamate, lactate, lactate + glutamate. Treatment with glutamate resulted in a statistically significant increase in dead cells compared to untreated cells. Data are presented as a mean ± SEM; n = 7; independent biological samples with 3 technical replicates. Statistical significance was determined using one-way ANOVA with Holm–Sidak test. * *p* < 0.05; *** *p* < 0.001. gluc: glucose; glut: glutamate; lact: lactate.

**Figure 5 biomolecules-14-00543-f005:**
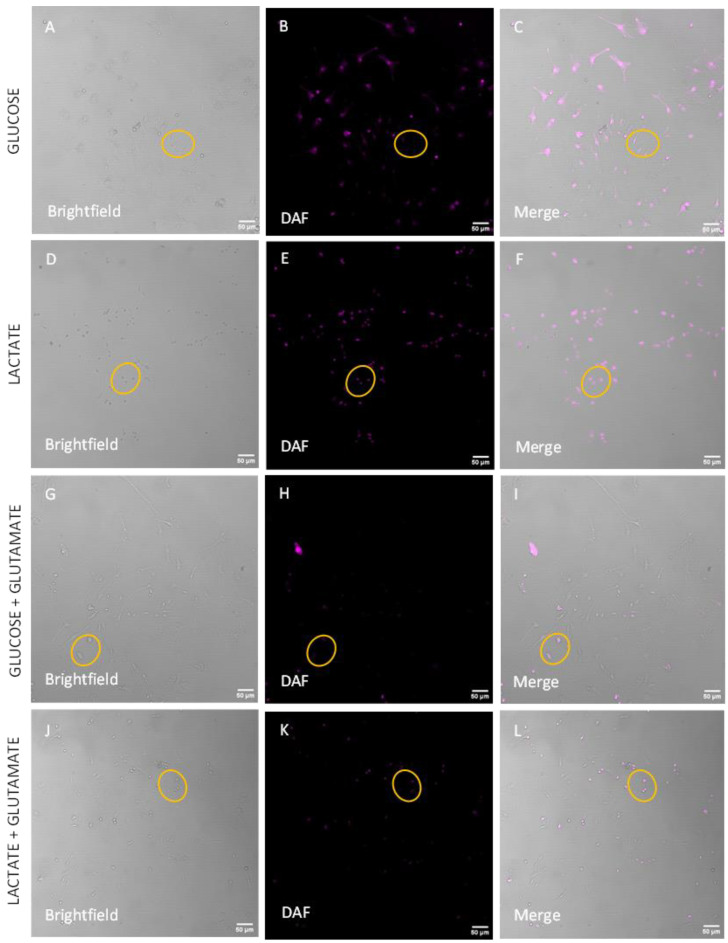
Intracellular NO in primary cortical cells isolated from Sprague-Dawley rats after adding different treatments: glucose (**A**–**C**), lactate (**D**–**F**), glucose + glutamate (**G**–**I**), and lactate + glutamate (**J**–**L**). The mean value of cells cultured with lactate increases the amount of endogenous NO in total cells (**M**,**N**) and neurons (**O**,**P**) in the absence (**M**,**O**) or presence (**N**,**P**) of glutamate. Glutamate did not induce any changes (**N**,**P**). The yellow circle is surrounding the neurons. Brightfield: (**A**,**D**,**G**,**J**). DAF: (**B**,**F**,**H**,**K**). Merge: (**C**,**F**,**I**,**L**). Data are presented as a mean ± SEM. n = 4 independent biological samples. Statistical significance was determined using ratio paired *t*-test. gluc: glucose; glut: glutamate; lact: lactate; DAF-FM: diaminofluorescein. Scale bar = 50 µm.

**Figure 6 biomolecules-14-00543-f006:**
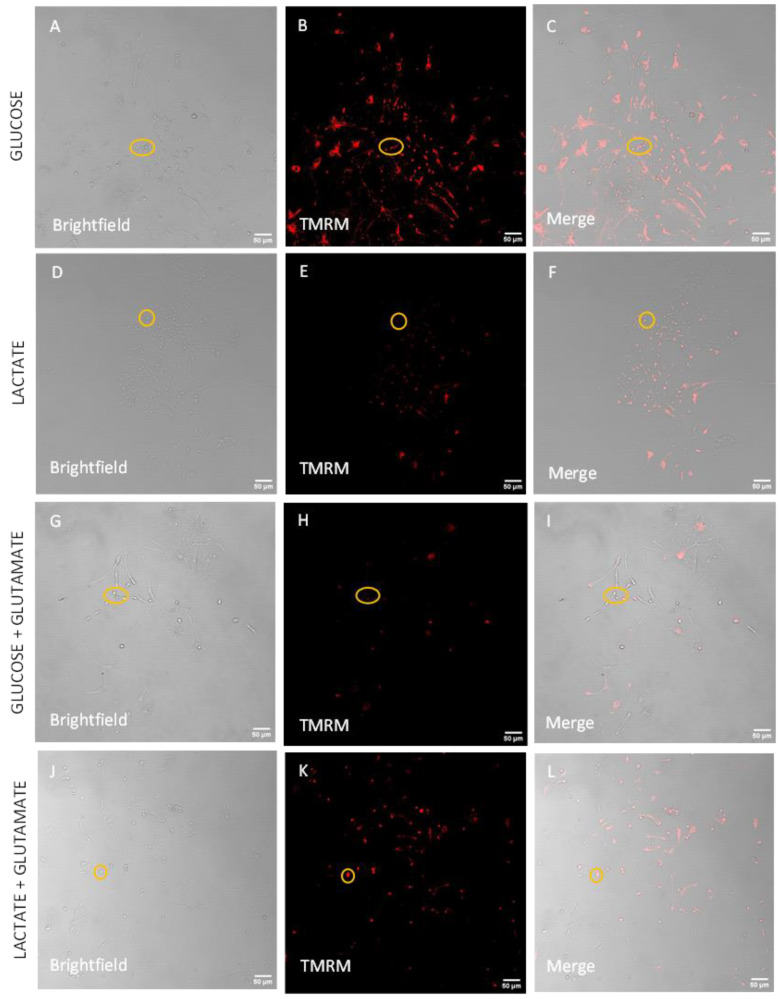
MMP (Δψ) of primary cortical cells isolated from Sprague-Dawley rats after adding different treatments: glucose (**A**–**C**), lactate (**D**–**F**), glucose + glutamate (**G**–**I**), lactate + glutamate (**J**–**L**). The cells cultured with glucose show a higher mean value for membrane potential in comparison to total cells (**M**) and significant changes in the neurons (**O**). The addition of glutamate increases the mean value of MMP in the presence of lactate of total cells (**N**) and leads to a significant change in the neurons (**P**). The yellow circle is surrounding the neurons. Brightfield: (**A**,**D**,**G**,**J**). TMRM: (**B**,**F**,**H**,**K**). Merge: (**C**,**F**,**I**,**L**). Data are presented as a mean ± SEM. n = 4 independent biological samples. Statistical significance was determined using Unpaired *t*-test (**M**,**O**) and Radio paired *t*-test (**N**,**P**). * *p* < 0.05. gluc: glucose; glut: glutamate; lact: lactate; TMRM: tetramethylrhodamine methyl ester. Scale bar = 50 µm.

**Figure 7 biomolecules-14-00543-f007:**
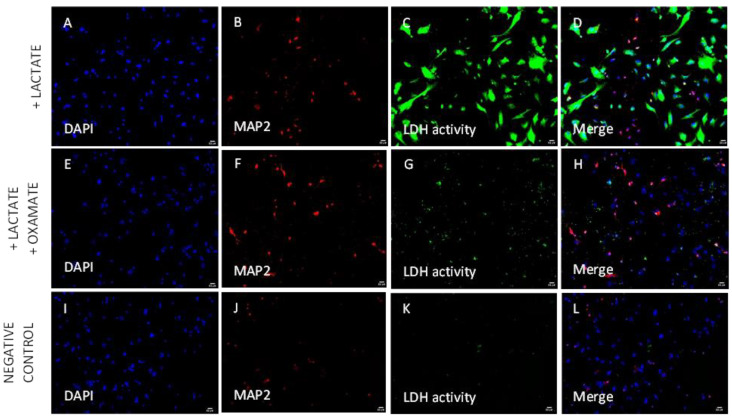
Lactate dehydrogenase activity in primary cortical cells isolated from Sprague-Dawley rats cultured in glucose medium was detected by nitro blue tetrazolium chloride in the presence of (**A**–**D**) lactate (substrate for LDH enzyme), (**E**–**H**) lactate + oxamate (inhibitor of LDH), or (**I**–**L**) non-treated cells (negative control). The signal corresponding to the LDH activity is only observed if lactate is added to the reaction; the activity of the enzyme is lower if it is inhibited by oxamate or without substrate. Red: MAP2 marker to detect microtubules in the neurons; blue: DAPI; green: LDH activity; MAP2: microtubule-associated protein 2; LDH: lactate dehydrogenase. Scale bar = 50 µm.

**Figure 8 biomolecules-14-00543-f008:**
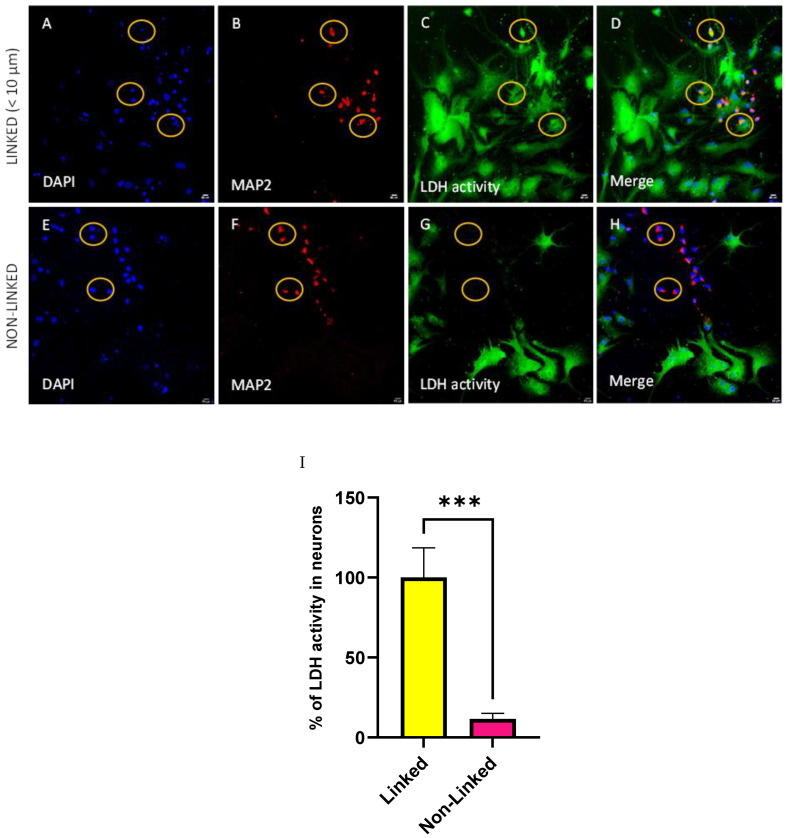
LDH activity in primary cortical cells isolated from Sprague-Dawley rats cultured in glucose medium was detected with nitro blue tetrazolium chloride in the presence of lactate (substrate for LDH enzyme). (**A**,**E**) nuclear staining with DAPI, (**B**,**F**) positive neurons for MAP2 marker, (**C**,**G**) positive cells for LDH activity, (**D**,**H**) merged images. Upper row (**A**–**D**): neurons which are in less than 10 µm of distance from the glial cells (linked neurons) present higher LDH activity in comparison to the cells that were located (lower row, (**E**–**H**)) at a greater distance than 10 µm (non-linked neurons) from glial cells. An example of each type of neuron is inside the yellow circle. (**I**) The quantification showed a higher LDH activity in linked neurons than in non-linked neurons. Red: MAP2 marker to detect microtubules in the neurons; blue: DAPI; green: LDH activity. Data are presented as mean ± SEM. n = 5–7. Statistical significance was determined using Unpaired *t*-test. *** *p* < 0.001. MAP2: Microtubule-associated protein 2; LDH: lactate dehydrogenase. Scale bar = 50 µm.

## Data Availability

The data are available upon request.

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
