# Peer review of "Interplay between Energy Supply and Glutamate Toxicity in the Primary Cortical Culture"

_biomolecules, 2024, doi:10.3390/biom14050543_

Round 1
Reviewer 1 Report
Comments and Suggestions for Authors
In the present study, the authors examined NO levels, mitochondrial membrane potential (MMP) and cell death after glutamate treatment in the presence of glucose or lactate. In addition, they examined difference of LDH activity between "linked" and "not-linked" neurons. Although this study includes interesting results, some of the conclusions lack enough evidence. Some of the experimental methods are insufficiently described. The following portions in the present manuscript need to be further addressed for publication in Biomolecules.
1. Title is somewhat vague but should be more specific.
2. Abstract is redundant but should be more compact.
3. In cell culture with glucose or lactate, the concentrations of glucose and lactate were 20 (or 25?) and 10 mM, respectively. The authors should explain the relevance of these concentrations in this study.
4. Fig. 4: The culture day of cells and concentrations of glucose, lactate and glutamate are not shown, although most of them were written in Materials and Methods (2.5). These are important information and should be shown also in Results.
In addition, when did glucose or lactate treatment start? From Fig. 4A, glucose or lactate treatment looks like starting before glutamate treatment? Is it right?
5. Fig. 5: How did the authors classify cells into neurons and glial cells? Although I suppose it is same to Fig. 6, it should be clearly shown.
6. Fig. 6: It is written that cells were classified into neurons and glial cells based on the size of the cell body (smaller or larger than 100 µm) in Materials and Methods (2.7). However, I think this classification is quite uncertain. To my knowledge, there are glial cells whose cell body is smaller than 100 µm.
7. Figs. 7 and 8: Were the cells used in these experiments cultured in the presence of glucose, not lactate? It should be shown.
8. Figs. 7 and 8: Although there are no data of glutamate treatment in Figs. 7 and 8, there are descriptions of glutamate treatment in Materials and Methods (2.4; lines 144 and 161). I cannot understand these descriptions.
9. Fig. 8: Although neurons were classified using immunostaining against MAP2, immunostaining against a glial marker was not performed. The authors should convincingly classify glial cells using immunostaining against a glial marker.
10. There are no experimental data concerning difference of responses to glutamate treatment between "linked" and "not-linked" neurons.
Comments on the Quality of English LanguageMinor editing of English language is required.
Reviewer 2 Report
Comments and Suggestions for Authors
Review: lactate makes the cells more susceptible to glutamate toxicity-2024
Overall, the manuscript is well written and explains the results clearly. The study addresses an important topic for researchers in using the experimental model of neuronal culture. Other researchers will benefit from the studies reported.
The aim of the study is well laid out …. “The aim of this study was to clarify whether incubation of primary cortical cells with glucose or lactate, as energy sources, affects the cellular viability and susceptibility to glutamate toxicity”
There are just some minor points which I think will aid future readers and some readers not specifically conducting research in this area.
Methods:
1. “only if the cells were positive for MAP2”
Should state in methods DAPI is used to label all cell types (glia and neurons)
-does this stain glia as well as neurons? Would the authors please state what cells are to be labeled with this antibody in the Methods.
In the RESULTS it states “The viability was quantified by counting MAP2 positive cells (Fig. 3H)” and figure legend “Red: MAP2-marker to detect microtubules in the neurons, blue: DAPI. Scale bar = 50 μm.”
So, one is to assume the MAP2 only labels neurons, but it would be good for a reader to know that explicitly.
2. For the sake of the readers can the authors state why these compounds are used and what their purpose is for ?
- methoxyphenamine methosulfate (Sigma-Aldrich),
- 5 mM sodium azide (Sigma-Aldrich),
- 5 mM nitroblue tetrazolium chloride (Sigma-Aldrich)
- 3 mM β-nicotinamide adenine dinucleotide hydrate (Sigma-Aldrich)
>>>>>>>>>>>>>>>>>>>>>>>>>>>>>>>>>
Potentially with glutamate added there is even a higher level of lactate produced by cells.
If lactate concentration was raised to the level if the increased production by the addition of glutamate, but not adding glutamate, would the death rate had gone up? Maybe at least address the possibility of the increase production of lactate by the addition of glutamate. This article below was not cited by the Authors.
“Lactate production is reportedly triggered by glutamate uptake, and independent of glutamate receptor activation.”
Cesar K, Hashemi P, Douhou A, Bonvento G, Boutelle MG, Walls AB, Lauritzen M. Glutamate receptor-dependent increments in lactate, glucose and oxygen metabolism evoked in rat cerebellum in vivo. J Physiol. 2008 Mar 1;586(5):1337-49. doi: 10.1113/jphysiol.2007.144154. Epub 2008 Jan 10. PMID: 18187464; PMCID: PMC2375663.
>>>>>>>>>>
It is known that lactate does add to Ca2+ influx and when glutamate is present would the authors care to address these earlier findings? This is one article of I few I found searching PubMed related to this matter. But this was not cited by the Authors.
“In conclusion, our results show that L-Lactate is involved in two distinct and independent pathways defined as NMDAR-mediated potentiation pathway (or NADH pathway) and a neuroprotective pathway (or Pyruvate/ATP pathway), the prevalence of each one depending on the strength of the glutamatergic stimulus.”
Jourdain, P., Rothenfusser, K., Ben-Adiba, C. et al. Dual action of L-Lactate on the activity of NR2B-containing NMDA receptors: from potentiation to neuroprotection. Sci Rep 8, 13472 (2018). https://doi.org/10.1038/s41598-018-31534-y
However, the authors did references other works in the Discussion section. So maybe the authors fine these studies sufficient to use to address the topic.
“We did not observe any difference in the cell death rate between the cells which were 366 incubating with glucose or lactate (Fig. 3). These data are in line with previous publications [39]. The authors quantified relative viability of primary cortical neurons treated 368 with 20 mmol/L of glucose and 10 mmol/L of lactate, and did not report any difference [39]. However, the cells incubated with lactate exhibited increased levels of NO and iNOS. Similar result was reported in SH-SY5Y neuronal cell line and in primary cortical astrocytes, following incubation with 10 mmol/L of lactate for 24 h [40]. The correlation be-372 tween nitric oxide production and LDH release has already been reported [41], suggesting the activation of NO mediated mechanism of cell death.”
>>>>>>>>>>>>
Results:
Line 276: Figure 5 legend “radio paired t-test” Do the authors mean “ratio paired t-test” ?
References:
Check journal format as some titles of papers are capitalized and others are not in the list shown.
>>>>>>>>>>>
Introduction, line 68: Just out of curiosity ...if a neuron that does not use glutamate as a neurotransmitter, but a neighboring neuron releases glutamate can the cell take up glutamine from a glia in close vicinity and use it as an energy source for conversion to alpha ketoglutarate in the TCA cycle? Thus, the neuron not using glutamate as a neurotransmitter, does not necessarily need to recycle glutamate only as a neurotransmitter. Is this correct?
Round 2
Reviewer 1 Report
Comments and Suggestions for Authors
The authors have fully responded to my comments. I have no further comments.